# Review of Cardiovascular Mock Circulatory Loop Designs and Applications

**DOI:** 10.3390/bioengineering12080851

**Published:** 2025-08-07

**Authors:** Victor K. Tsui, Daniel Ewert

**Affiliations:** Department of Biomedical Engineering, University of North Dakota, Grand Forks, ND 58202, USA; daniel.ewert@und.edu

**Keywords:** cardiovascular, mock circulatory loop, in vitro test, regulatory engineering

## Abstract

Cardiovascular diseases remain a leading cause of mortality in the United States, driving the need for advanced cardiovascular devices and pharmaceuticals. Mock Circulatory Loops (MCLs) have emerged as essential tools for in vitro testing, replicating pulsatile pressure and flow to simulate various physiological and pathological conditions. While many studies focus on custom MCL designs tailored to specific applications, few have systematically reviewed their use in device testing, and none have assessed their broader utility across diverse biomedical domains. This comprehensive review categorizes MCL designs into three types: mechanical, computational, and hybrid. Applications are classified into four major areas: Cardiovascular Devices Testing, Clinical Training and Education, Hemodynamics and Blood Flow Studies, and Disease Modeling. Most existing MCLs are complex, highly specialized, and difficult to reproduce, highlighting the need for simplified, standardized, and programmable hybrid systems. Improved validation and waveform fidelity—particularly through incorporation of the dicrotic notch and other waveform parameters—are critical for advancing MCL reliability. Furthermore, integration of machine learning and artificial intelligence holds significant promise for enhancing waveform analysis, diagnostics, predictive modeling, and personalized care. In conclusion, the development of MCLs should prioritize standardization, simplification, and broader accessibility to expand their impact across biomedical research and clinical translation.

## 1. Introduction

With changing lifestyles and an aging population, the prevalence of cardiovascular diseases is significantly increasing. Cardiovascular disease remains one of the leading causes of death in the United States, creating a high demand for cardiovascular medical devices and pharmaceuticals. To address this need, scientists and engineers have developed Mock Circulatory Loops (MCLs) to assist in testing these medical devices prior to clinical trials. An MCL is an in vitro system that provides pulsatile pressure and flow, simulating various physiological and pathological parameters of the human circulatory system. Most MCL setups replicate the parameters and mechanisms of the heart, valves, and compliance chambers. As a result, MCLs have become essential tools in medical research and engineering.

While numerous studies describe the design and use of MCLs, most focus on customized systems developed for very specific applications. Literature often emphasizes individual configurations rather than broader comparative analyses. MCL designs are categorized into three primary types: mechanical, computational, and hybrid models. This review not only examines different MCL designs but also explores various MCL applications. Specifically, MCL applications are classified into the following four major areas:Cardiovascular Devices Testing;Clinical Training and Education;Hemodynamics and Blood Flow Studies;Disease Modeling.

Recent advancements in MCLs have significantly improved both physiological accuracy and waveform fidelity [1]. The integration of advanced control systems—such as programmable microcontrollers and refined pump control mechanisms—has enhanced the dynamic responsiveness and precision of MCLs [2,3]. Moreover, the emergence of hybrid models that integrate computational and mechanical components has greatly increased the flexibility of MCL waveform simulations, enabling more realistic replication of diverse cardiovascular conditions [4]. These technological improvements have expanded the scope of MCL applications. While traditionally used primarily for a particular cardiovascular device testing [5], modern MCLs are now being applied to a wider range of research areas. MCLs are also employed in Clinical Training and Education, providing medical students and clinicians with a hands-on, risk-free platform to study cardiovascular dynamics [6]. Furthermore, MCLs are increasingly used in Hemodynamics and Blood Flow Studies, enabling detailed investigation of physiological and pathological flow conditions [7]. Notably, recent studies emphasize the growing role of MCLs in cardiovascular waveform analysis for disease evaluation and physiological modeling [8,9,10]. This trend highlights the potential for MCLs to serve as comprehensive platforms for both device innovation in device design and translational cardiovascular research [11].

However, most existing MCL systems are tailored for specific applications, limiting their flexibility and scalability across broader research and clinical applications. This comprehensive review explores the design and evolution of MCLs, examining studies that have utilized MCLs to simulate cardiovascular systems, their technological advancements, and discussing their applications in research and clinical practice. There is a technical need for a simplified, modular, reconfigurable, and standardized MCL platform [12] that can be adapted to a wide range of applications. One particularly critical application is cardiovascular waveform analysis, which plays a key role in assessing hemodynamic performance and cardiovascular health. Further research is needed to enhance the evaluation, characterization, and prediction of waveform features such as pressure, flow, and pulse waveforms to achieve more accurate diagnostics, real-time monitoring, and better patient outcomes.

## 2. Methodology

This literature review focuses on the applications of MCLs in existing research and clinical practice. This review paper covered various types and designs of MCLs, ranging from simple to advanced systems that mimic systemic and/or pulmonary circulation for different applications. The primary sources of this review were papers and patents published from the following databases:Google Scholar;PubMed;IEEE Xplore Digital Library;Web of Science;Google Patents.

These databases were chosen for their comprehensive coverage of the relevant scientific and engineering literature. The search strategy included keywords such as “mock circulatory loop”, “cardiovascular simulation”, “in vitro cardiovascular models”, and “cardiovascular device testing”. Studies were selected based on their relevance to the topic, the novelty of their findings, and their contributions to the field. This literature review includes related patent searches for MCLs. This patent search focuses on the unique design of the MCLs and any special applications. This literature review will summarize three MCL design models and categorize four different applications of MCLs, as shown in Figure 1: Flowchart of this literature review.

## 3. Design of Mock Circulatory Loops

The historical development of MCLs dates back to 1952, when McMillan et al. [13] constructed a basic circulatory loop using color cinematography to study the movements of aortic and pulmonary valves. This initial design featured an electronic valve that intermittently opened and closed to produce pulsatile flow. Subsequently, in the late 1960s, Liotta et al. [14] documented the first generation of MCLs, marking a significant advancement in cardiovascular simulation. A review of patents reveals early innovations in MCL design. Notably, in 1969, Willem Kolff and Stephen Jacobsen patented a mock circulation system comprising enclosed chambers representing arteries, veins, and organs, with artificial ventricles to pump fluid, thereby developing circulation within the system [15]. In 2006, Jeremy Low and Von Huben patented a mock circulatory apparatus designed for testing cardiovascular devices, highlighting the ongoing evolution of MCL technology [16]. While recent patents specifically focusing on MCL designs are limited, there has been a surge in patents related to the application of MCLs in testing various cardiovascular devices, such as heart valves and ventricular assist devices (VADs).

A standard MCL typically comprises the following components, as shown in Figure 2: Schematic of MCL:Pulmonary Circuit: Simulates the blood flow between the heart and lungs.Heart: Includes representations of the left and right atria and ventricles, often equipped with the mechanisms to mimic cardiac contractions.Systematic Circuit: Represents the circulation of blood to the rest of the body, including arterial compliance chambers and systemic resistances.

In simplified MCLs, the design may exclude the right atrium and ventricle, with the flow returning directly to the left atrium after passing through systemic resistance. The evolution of MCLs reflects a continuous effort to enhance the fidelity of cardiovascular simulations, thereby improving the development and testing of medical devices and therapeutic interventions.

MCLs can be broadly classified into the following three categories [5]:Mechanical (Hydraulic) MCLs: These systems utilize physical components to replicate cardiovascular functions. They are known for their high repeatability but may lack flexibility in simulating autoregulatory responses.Computational (Numerical) MCLs (cMCLs): These models employ computational methods to simulate cardiovascular dynamics, offering flexibility and programmability. However, they lack the physical properties necessary for in vitro testing of actual cardiovascular devices.Hybrid MCLs (hMCLs): Combining elements of the mechanical and computational models, hybrid MCLs aim to leverage the advantages of each approach to provide more comprehensive simulations.

### 3.1. Mechanical Mock Circulatory Loops

Typical mechanical MCLs consist of components that simulate the heart chambers, valves, compliance chambers, and vascular resistance, as illustrated in Figure 3. This configuration represents a basic design focusing on systemic circulation only. In this setup, mechanical pumps—such as centrifugal or peristaltic pumps—are used to simulate the function of the heart ventricles, generating pulsatile flow that mimics the contraction and relaxation phases of the cardiac cycle. These pumps move a working fluid (usually water or a blood analog) through the system, replicating the output of the left and right ventricles. Check valves are strategically placed in the circuit to serve as mitral valves and aortic valves, ensuring unidirectional flow and preventing backflow during the pumping cycle. To replicate the atriums, fluid containers or flexible bags are used, functioning as temporary reservoirs that hold the fluid before it enters the pump. Systemic compliance is simulated using an air-filled chamber or vertical water column, which dampens pressure and flow fluctuations, much like the elasticity of arteries. Finally, systemic vascular resistance is controlled via flow regulators or resistive tubing, allowing users to tune pressure and flow to the desired physiological conditions. All components are connected with flexible, transparent tubing, enabling visual observation, easy configuration changes, and mimicking blood vessel elastance. This modular and mechanical design provides a practical and repeatable platform for cardiovascular device testing, educational demonstrations, and basic hemodynamic studies.

Kolff et al. [17] filed another patent in 1994, an educational tool featuring manually squeezing the artificial ventricles, allowing users to visualize and understand cardiac mechanics. Gurdin et al. [18] and Leonhardt et al. [19] filed patents for an artificial heart pump as part of MCL components. Vozenilek et al. [20] built an advanced MCL that incorporates four pumps and ultrasound sensors to generate and capture the hemodynamic waveform. This system used an anatomical unit, pumps, and solenoid valves to simulate blood flow. Pickard et al. [21] and Gregory et al. [22] filed a unique design of MCLs for in vitro testing of heart valve and cardiovascular devices.

Khienwad et al. [23] built a unique design of a Frank–Starling physiology MCL, which utilizes a rotary pump and tall water columns to replicate the Frank–Starling mechanism, demonstrating the relationship between ventricular filling and stroke volume. Frank–Starling principle states that the stroke volume of the heart increases in response to an increase in the volume of blood filling the heart (end-diastolic volume), up to a physiological limit [24,25]. Gregory et al. [26] incorporated a proportional controller based on ventricular end-diastolic volume to adjust driving pressure. Another fundamental physiological principle is the Windkessel Model. It describes arterial compliance and systemic vascular resistance to smooth out pulsatile pressure and flow from the heart to maintain steady perfusion [27]. Some advanced MCLs include Baroreflex and autoregulation feedback control to regulate blood pressure. Baroreceptors in the body adjust heart rate and vascular resistance in response to pressure changes. Mushi et al. [28] presented an MCL integrated with a baroreceptor reflex.

### 3.2. Computational Mock Circulatory Loops

cMCLs are software-based models that simulate cardiovascular dynamics, enabling researchers to study various physiological and pathological conditions without physical prototypes. These models utilize mathematical representations to emulate blood flow, pressure, and other hemodynamic parameters, offering flexibility and adaptability in cardiovascular research. Several computational tools and platforms are employed to develop cMCLs. MATLAB, Simulink, ANSYS, COMSOL, and Python are widely used for modeling, simulating, and analyzing cardiovascular models that can replicate human pressure and flow waveforms. A notable example is the work by De Lazzari et al. [29], who developed an updated version of CARDIOSIM©, an Italian software platform designed for cardiovascular system simulation. CARDIOSIM© is distinguished by its patient-specific approach, enabling the customization of simulations to match individual physiological profiles for enhanced clinical relevance.

Typical cMCLs are composed of numerical models that represent key components of the cardiovascular system, including the aortic valve, left ventricle, aorta, venous system, and systemic resistance, as illustrated in Figure 4. These models are governed by time-dependent differential equations that simulate the dynamics of flow (Q) and pressure (P) across the system under varying physiological conditions. The computational framework uses parameters such as resistance (R), compliance (C), and elastance (E) to mimic the mechanical behavior of each model. By adjusting these parameters over time (t), the model can simulate realistic cardiac cycles, pressure–volume relationships, and hemodynamic response to interventions or pathological conditions. cMCLs offer a flexible and precise approach for studying cardiovascular function and testing medical devices in silico.

Quarteroni et al. [30] studied the use of mathematical modeling and numerical simulation in cMCLs, highlighting the value of computational models in replicating complex physiological conditions. cMCLs can be tailored to the specific application, allowing researchers to simulate more advanced physiological behaviors such as vascular autoregulatory and multi-organ interactions, which are difficult to achieve in purely mechanical models. Similarly to the mechanical MCLs, computational models can also use fluid mechanical simulation, such as pumps, regulators, reservoirs, and check valves, to model blood flow dynamics [31]. Gregory et al. [32] built a mathematical model of a cMCL in a MATLAB/Simulink environment. Taylor et al. [33] created a cMCL compliance chamber proportional integral control using Simulink/Simscape hydraulic mechanical toolboxes. Additionally, researchers can also use electrical circuit analogs such as resistance, current, voltage, capacitance, and inductance to represent blood flow resistance, flow rate, pressure, vascular compliance, and inertia effects, respectively [34]. Lumped-parameter models are widely used in cMCL with the CellML mark-up language [35]. Korakianitis et al. [36] modeled the heart using variable or constant elastance functions and included four valves to control blood flow direction. In their model, the change in left ventricular volume over time (dV_lv/dt) is defined as the difference in flow between the mitral and aortic valves (Q_mi − Q_ao).

A simplified cMCL using a time-varying elastance model for the left ventricle and a systemic resistance loop was implemented in MATLAB R2024a. This model simulates key cardiovascular parameters and captures the dynamic interaction between pressure and flow within the systemic circulation. The resulting simulations of left ventricular pressure, aortic pressure, and valve flows are illustrated in Figure 5. These simulated waveforms can be utilized for detailed cardiovascular waveform analysis and Disease Modeling, particularly in conditions such as heart failure and hypertension. Additionally, this cMCL framework provides a versatile and reproducible platform for the evaluation of cardiovascular devices, including prosthetic heart valves (PHVs), VADs, and stents. By providing physiologically relevant boundary conditions and pressure–flow relationships, the model supports both design optimization and performance validation in a controlled simulation environment.

### 3.3. Hybrid Mock Circulatory Loops

hMCLs integrate mechanical components with computational models to enhance the simulation of physiological responses. This combination leverages the physical accuracy of mechanical systems and the adaptability of computational simulations, allowing for more precise replication of cardiovascular conditions. hMCLs consist of a mechanical subsystem and a computational subsystem. Mechanical subsystem includes physical components similar to Figure 3; however, typical hMCLs will consist of a communication or controlling system as a computational subsystem. This computational subsystem can read real-time pressure–flow data from sensors and compute updated values of resistance, compliance, flow regulator, or pump output. These systems are designed to overcome the limitations of purely mechanical or purely computational loops by combining the realistic fluid dynamics of physical systems with the adaptability and precision of mathematical modeling.

A notable example of an hMCL is the system developed by Bardi et al. [37], which integrates real-time control valves and 3D-printed phantom aortic multi-branches to investigate aortic phantom hemodynamics. Cuenca-Navalon et al. [38] introduced an hMCL with computer-controlled hydraulic compliance chambers, enhancing the system’s ability to replicate dynamic cardiovascular conditions. This design improves the accuracy of simulations, particularly in replicating patient-specific scenarios. Li et al. [39] developed a comprehensive hMCL capable of generating detailed waveform descriptions for patient-specific applications. This system’s adaptability allows researchers to simulate a wide range of cardiovascular conditions, facilitating personalized device testing and development. Another innovative design by Timms et al. [40] featured a compact hMCL tailored for in vitro testing of cardiovascular devices. This system mimics several hemodynamic parameters, including arterial compliance and systemic and pulmonary resistances, offering a versatile platform for device evaluation. Iscan et al. [41] built an hMCL with active capacitance as multiple compliance chambers. Taylor et al. [42] used Simulink and Simscape to build a fully automated hMCL.

### 3.4. Mock Circulatory Loops Design Summary

The comparison of the three main types of MCL models is summarized in Table 1.

The International Organization for Standardization (ISO) published ISO 14708-5:2020, titled “Implants for surgery—Active implantable medical devices—Part 5: Circulatory Support Devices.” [43]. This standard specifies requirements for the safety and performance of active implantable circulatory support devices, including type tests, animal studies, and clinical evaluation requirements. Imachi et al. [44] studied this ISO standard, which is applicable to devices commonly referred to as active implantable medical devices. The U.S. Food and Drug Administration (FDA) has developed an MCL designed to simulate peripheral (radial) blood pressure waveforms [12]. This MCL is intended for non-clinical characterization of pressure-based cardiac output monitoring systems and can simulate three hemodynamic states: cardiogenic shock, normovolemic state, and hyperdynamic state. Several companies offer commercially available MCL systems and pulsatile pumps. Vivitro Labs provides circulatory support device testing services, including pump performance, fluid dynamic analysis, cavitation observation, and system characterization. Their testing simulates the effects of changes in system performance on the “patient” (e.g., pulse duplicator) and vice versa. BDC Laboratories offers custom-built MCLs tailored to specific testing requirements, providing solutions for cardiovascular device testing. Biopac Systems and Harvard Apparatus supply general-purpose MCLs suitable for educational purposes, facilitating the study of cardiovascular physiology and device interactions. Transonic manufactures mechanical circulatory support-related medical devices and has published a flow measurement guide for industry bioengineers: Mock Circulatory Loops [45]. These commercially available systems and components enable researchers and educators to conduct in vitro testing of cardiovascular devices, study hemodynamic responses, and simulate various physiological conditions in a controlled environment.

## 4. Applications of Mock Circulatory Loops

This literature review aims to explore the various applications of MCLs in different areas. MCL applications are classified into the following four categories:Cardiovascular Devices Testing;Clinical Training and Education;Hemodynamics and Blood Flow Studies;Disease Modeling.

### 4.1. Cardiovascular Devices Testing

MCLs are essential in vitro systems designed to replicate human cardiovascular physiology, providing a controlled environment for testing and optimizing cardiovascular devices before proceeding to in vivo studies. They allow researchers to simulate various hemodynamic and pathological conditions, ensuring device functionality, safety, and durability. Among cardiovascular devices, total artificial hearts (TAHs) or VADs are extensively tested by MCLs [46,47,48] for functionality, safety, and durability evaluation. Xu et al. [5] demonstrated that MCLs can be used to test TAHs, Cardiovascular Assist Devices, VADs, Intra-Aortic Ballon Pump (IABP), or a pump-type device that pumps blood in preclinical stages. For instance, D’Souza et al. [10] used MCL to simulate varying degrees of heart failure, allowing for the optimization of device flow rates and patient-specific conditions.

Mechanical and PHVs are another common type of cardiovascular devices tested by MCLs. A typical setup involves integrating the PHV into the MCL and placing pressure sensors upstream and downstream of the valve, as illustrated in Figure 6. This configuration enables researchers to adjust hemodynamic conditions within the MCL and monitor resulting changes in pressure and/or flow, providing valuable data for evaluating the performance of PHVs under various physiological scenarios.

Researchers employ MCLs to investigate the motion and hemodynamic performance of PHVs under diverse conditions. Lin et al. [49] connected a PHV to an MCL to observe and quantify gas bubble formation on mechanical heart valves. Similarly, Feng et al. [50] used high-speed cameras within an MCL to study PHV dynamics, providing insights into valve behavior during cardiac cycles. Walker et al. [51] also used a novel test chamber for mechanical heart valve performance. Vismara et al. [52] used a novel approach to study the aortic valve performance.

Life-sized MCLs serve as prototypes for short-term peripheral mechanical support (extracorporeal life support, ECLS). Gehron et al. [53] developed an advanced MCL that, after successful validation, could function as an ECLS system, offering a platform for testing and refining these critical devices.

MCLs facilitate the assessment of various cardiovascular devices, including prosthetic aortic grafts and intravascular bioartificial organs. Ferrari et al. [54] utilized an MCL to study the dynamic behavior of a prosthetic aortic graft, while Moyer et al. [55] explored peripheral intravascular bioartificial organs within an MCL setup. Additionally, computational models integrated with MCLs have been employed by Wang et al. [56] to analyze guidewire hemodynamics, enhancing the development of minimally invasive procedures. Many other researchers are using MCL to test their innovative devices during development, such as a continuous blood pressure monitor, abdominal aortic aneurysm graft leak monitor, and more.

### 4.2. Clinical Training and Education

MCLs have become invaluable tools in clinical training and surgical simulations, offering realistic platforms for medical education and skill development. MCLs are extensively used to simulate various cardiovascular conditions, providing healthcare professionals with hands-on experience in a controlled environment. For instance, Pantalos et al. [57] developed a cardiovascular simulator tailored for pediatric research and training, enabling practitioners to better understand and manage pediatric cardiac conditions. Similarly, Carson et al. [58] and Telyshev et al. [6] utilized pneumatic MCL systems to replicate diverse heart conditions, facilitating comprehensive training in heart failure management. These systems allow clinicians to practice and refine their skills without posing risks to actual patients.

MCLs have also been instrumental in training for catheterization procedures and endovascular surgeries. Liang et al. [59] constructed a simulator using silicone phantoms integrated with an MCL to provide realistic catheterization training and to assess endovascular surgical skills. This setup offers trainees a tactile and visual representation of human vasculature, enhancing their procedural proficiency. Additionally, Johnson et al. [60] developed an affordable MCL designed for cardiac catheterization procedures using a pulmonary artery catheter, making such training more accessible.

In regions where traditional medicine practices, such as palpation, are prevalent, MCLs serve as effective training tools. The practice of palpation is discussed in the first chapter of Physical Examination of the Heart and Circulation, a book written by Dr. Perloff [61]. Jeong et al. [62] employed MCLs to simulate cardiovascular physiology, aiding medical practitioners in honing their diagnostic skills through palpation. This application underscores the versatility of MCLs in accommodating various diagnostic techniques across different medical traditions. A similar setup is illustrated in Figure 7, where a mechanical MCL is connected to a silicone-molded phantom arm. Silicone tubing is embedded within the phantom to mimic the radial artery, allowing for palpation training under controlled hemodynamic conditions.

MCLs are widely used for clinician training in Extracorporeal Life Support (ECLS) [63]. Geier et al. [64] and Mahmoud et al. [65] utilized MCLs to study and train clinicians in cannulation techniques during Extracorporeal Membrane Oxygenation (ECMO), a critical component of ECLS. These training modules ensure that healthcare providers are well-prepared to perform complex procedures, ultimately improving patient outcomes.

### 4.3. Hemodynamics and Blood Flow Studies

MCLs are invaluable tools for in vitro studies of hemodynamics, enabling researchers to simulate and analyze cardiovascular conditions such as hypertension, atherosclerosis, and aneurysms. By replicating physiological blood flow and pressure conditions, MCLs provide a controlled environment to investigate the complexities of these diseases without the immediate need for animal or human subjects. By mimicking the cardiovascular system’s dynamic behaviors, MCLs enable the simulation of various pathological conditions—such as hypertension (by increasing systemic resistance), aortic stenosis (by introducing narrowed valve elements), and heart failure (by reducing pump output or ventricular compliance). These controlled simulations allow researchers to quantify changes in flow patterns, pressure–volume relationships, and shear stress under specific hemodynamic conditions.

Packy et al. [66] investigated the hemodynamic profiles generated by MCLs across three physiological states: normovolemia, cardiogenic shock, and hyperdynamic circulation. Their experimental setup, shown in Figure 8, incorporated multiple pressure and flow sensors for comprehensive monitoring. In another study, Franchini et al. [67] conducted specialized studies on the viscoelasticity of human descending thoracic aortas within an MCL. Barua et al. [7] utilized MCLs to examine cardiac hemodynamics and renal flow concerning intra-aortic balloon pump (IABP) and a novel intra-aortic entrainment pump (IAEP). Additionally, Schwarzel et al. [68] employed MCLs to study the Extracorporeal Carbon Dioxide Removal (ECCO2R) rate without relying on animal models. Stephens et al. [69] and Rozencwaig et al. [70] utilized MCLs to investigate Venoarterial ECMO, while Mi et al. [71] incorporated sound sensors in MCLs to study the pulmonary artery pressure and heart valve vibration sounds. Papaioannou et al. [72] performed studies on ventricular dysfunction on arterial compliance using MCLs, and Hong et al. [73] combined commercial pulsatile pumps and 3D-printed aortoiliac arterial models connected to MCLs to characterize hemodynamics in human systemic arteries with stenosis. Koenig et al. [74] explored pressure and volume responses to continuous and pulsatile ventricular assist in MCLs. Li et al. [75] conducted a hemodynamic evaluation of a patient-specific aortic phantom integrated into an MCL, utilizing pressure sensors and ultrasonic flow sensors to assess flow dynamics and pressure distributions.

### 4.4. Disease Modeling

MCLs are powerful tools for cardiovascular Disease Modeling, enabling researchers to replicate specific pathological conditions and study their hemodynamic impact in a controlled and repeatable environment. Common cardiovascular diseases—such as heart failure, hypertension, aortic stenosis, aneurysms, and stroke—can be modeled in MCLs by adjusting parameters like loop resistance, tubing compliance, pump output, and valve functionality.

MCLs have been utilized to simulate conditions such as cardiogenic shock. For example, Contarino et al. [76] validated a Multiscale computational model using an MCL to simulate cardiogenic shock. Taylor et al. [77] employed Simulink Simscape to mimic peripheral resistance devices, and Schroeder et al. [78] utilized MATLAB for modeling cardiovascular loops. MCLs have also been applied in waveform analysis and diagnostics, sometimes incorporating machine learning techniques for enhanced pattern recognition and classification. As illustrated in Figure 9, a representative brachial artery pressure waveform—sourced from the database created by Willemet et al. [79]—is analyzed using MATLAB R2024a to extract key morphological features. These include the systolic peak, dicrotic notch, diastolic low, as well as the anacrotic limb (upstroke between the diastolic onset and systolic peak) and dicrotic limb (downstroke following the systolic peak). These landmarks provide critical insights into arterial stiffness, left ventricular contractility, and vascular compliance. Such quantitative analysis forms a critical foundation for diagnostic algorithms and machine learning techniques aimed at detecting abnormalities in vascular tone or cardiac function.

The Mean Arterial Pressure (MAP) is a critical parameter representing average blood pressure, calculated as diastolic pressure plus one-third of the difference between systolic and diastolic pressures [80,81]. Resources like the Thoracic Key website [82] provide extensive information on waveform analysis. Fan et al. [83] offered an online resource on pulse wave analysis, and researchers such as Jozwiak et al. [84] and Geng et al. [85] have conducted studies on pressure waveform analysis. Wang et al. [86] applied Gaussian and least squares fittings for waveform analysis. Baturalp et al. [87] researched valvular heart disease, specifically aortic stenosis, using systemic MCLs, while Liang et al. [88] employed Multiscale modeling of MCLs to study aortic valvular and arterial stenoses. Loredo et al. [89] developed a cMCL to investigate hemodynamics related to cerebral and abdominal aortic aneurysms.

## 5. Summary and Discussion

Based on this literature review, there are many applications for MCLs [48,90]. Four major MCL applications are summarized in Table 2. The majority of the MCL applications involve testing commonly used cardiovascular devices [91], such as VADs and PHVs. MCLs have become indispensable in cardiovascular research, from heart valve testing to VAD optimization. However, there is limited research on the application of MCLs in pediatrics, and their use for clinical training and teaching is also very limited. Hong et al. [92] recently conducted a comprehensive review of the MCLs applications, with a particular focus on their roles in surgical planning and Disease Modeling. In this research, we extensively examine various MCL designs and review different types of applications. When multiple papers address very similar applications, only representative studies are cited. Although the applications may be similar, the MCL designs often differ. Given their distinct characteristics, different types of MCL models are often more suitable for particular applications. Typical applications for the three main types of MCL models are summarized in Table 3.

When designing an MCL, challenges arise in replicating complex human conditions such as arrhythmia or congenital heart defects. hMCLs represent the latest trend and new generation [93] because they are more flexible in mimicking various hemodynamic and physiological conditions. Although there is an ISO standard for the general safety and use of MCLs, there is no specific standard for their design. Consequently, MCLs remain very application-specific or individually designed. Because no single standardized MCL is widely used for developing and evaluating cardiovascular devices, the capabilities and complexity of MCLs tend to vary among laboratories based on stakeholder needs, device types, and applications. For example, the FDA published a poster presentation titled “Use of Mock Circulatory Loops for Standardized Performance Testing of Mechanical Circulatory Support Devices: An Interlaboratory Study” [94]. However, there remains a need for a design standard or guideline for MCLs.

For instance, MCLs used for developing therapeutic cardiovascular devices (e.g., heart valves, vascular conduits and grafts, and catheters [54]) may vary from those used for evaluating novel diagnostic tools (e.g., for detecting vascular and valvular stenosis, aneurysms, and congenital heart defects). Despite the extensive use of MCLs for research and development, they are often underutilized in regulatory submissions of mechanical circulatory support (MCS) devices, particularly in simulating disease states such as heart failure conditions [10].

Cardiovascular waveform analysis [66,82] is another area that requires further development. With advances in computational modeling, researchers can develop applications for waveform analysis and diagnostics. As discussed in the context of Disease Modeling, feature-based analysis is a key method in waveform evaluation. Combined with machine learning algorithms, this approach can enable the classification of vascular types and the detection of abnormalities in cardiac function. Furthermore, such feature-based analysis may be valuable for validating the physiological fidelity of MCLs. Additionally, the integration of MCLs into personalized medicine and in vitro drug testing is very limited [95].

## 6. Conclusions and Future Directions

With the rise in cardiovascular disease, the demand for cardiovascular devices has significantly increased. MCLs play an important role in the development of cardiovascular devices and pharmaceuticals. Despite their importance, the applications and market value of MCLs remain underestimated. Mechanical and computational models alone may be replaced by hybrid models for improved accuracy and versatility. With the 3D printing technologies, anatomical models such as abdominal aortic aneurysms, arteries, heart valves, and more [96] with different compliance/elastance can be produced and used in MCLs [11]. Researchers should focus on developing flexible, cost-effective, and standardized MCL designs. However, since each MCL design is unique, most are complicated and challenging to construct. There is a need for a simpler, more flexible MCL design, along with standardized guidelines to ensure consistency and usability. A flexible MCL design with different configurations and programs can be used for different applications. Hybrid MCLs integrated with Wavelet or Fourier Transform techniques can be used to control and program pump pulse-width modulation (PWM) to accurately simulate desired cardiovascular output waveforms. These configurations enable the decomposition of original or target waveforms using Wavelet or Fourier transforms, allowing for more precise waveform replication tailored to various applications.

MCL’s primary applications include testing cardiovascular devices, training and teaching, studying hemodynamics and blood flow, and modeling the cardiovascular system and disease. Another significant application of MCLs is in waveform analysis and diagnostics, where the fidelity of the waveform needs to be better quantified beyond the current reliance on systolic, diastolic, and MAP [97]. In most of the MCL output waveforms, the dichroic notch is missing. Moreover, advanced waveform and pulse wave analysis can aid in disease prediction and prevention, such as pulse pressure and MAP in relation to ischemic stroke [98]. Cardiovascular waveform analysis is a critical tool for understanding hemodynamics and assessing cardiovascular health. To advance this field, more sophisticated and comprehensive waveform analysis indices should be developed and validated. Additional important parameters, such as the dicrotic notch and anacrotic limb, should also be incorporated. Further research is needed to improve the evaluation, characterization, and prediction of waveform components to enhance diagnostic accuracy and patient outcomes. Additionally, integrating machine learning and artificial intelligence (AI) into waveform analysis presents promise for enhancing predictive capabilities and early disease detection.

## Figures and Tables

**Figure 1 bioengineering-12-00851-f001:**
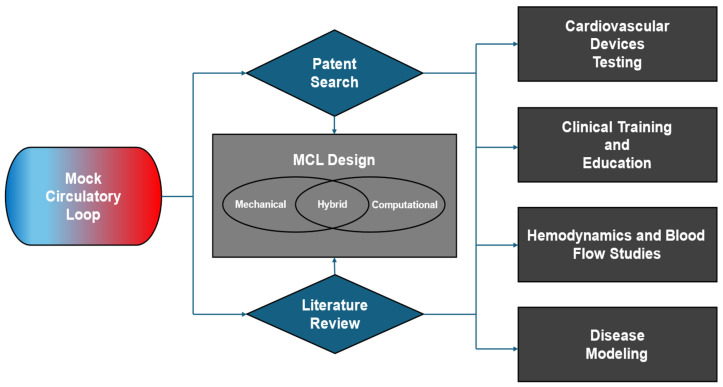
Flowchart illustrates the structure of this literature review on Mock Circulatory Loops. The flowchart visually represents the organization of this review from left to right. The MCL Design section in the center categorizes MCLs into three models. On the right, MCL applications are classified into four major categories.

**Figure 2 bioengineering-12-00851-f002:**
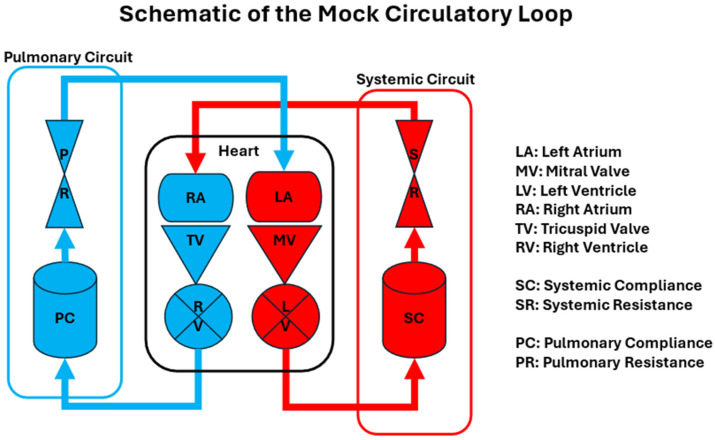
Schematic of a typical Mechanical Mock Circulatory Loop design. Mechanical pumps are normally used to pump fluid (blood analog) that mimics the function of the left and right ventricles. Check valves serve as mitral and tricuspid valves, while a reservoir or bag represents the left or right atrium. A pressurized chamber or water column functions as the compliance chamber. Tubing and flow regulators are utilized to adjust the resistance of the systemic or pulmonary circulation.

**Figure 3 bioengineering-12-00851-f003:**
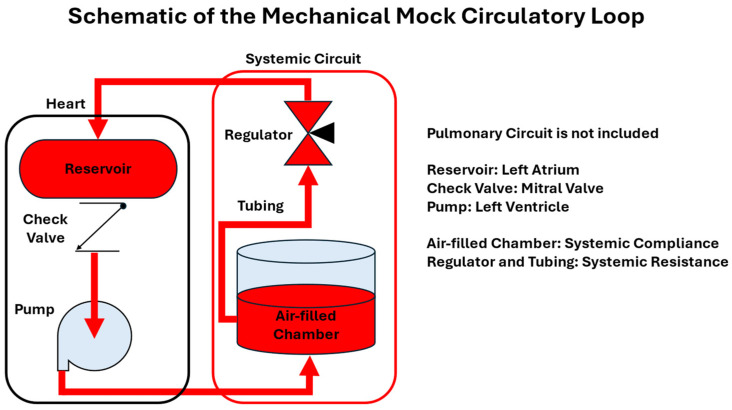
Schematic of a typical Mechanical Mock Circulatory Loop design.

**Figure 4 bioengineering-12-00851-f004:**
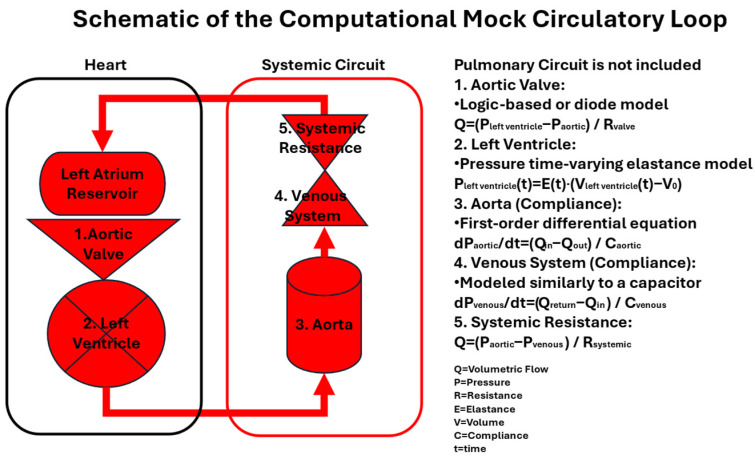
Schematic of a typical Computational Mock Circulatory Loop design.

**Figure 5 bioengineering-12-00851-f005:**
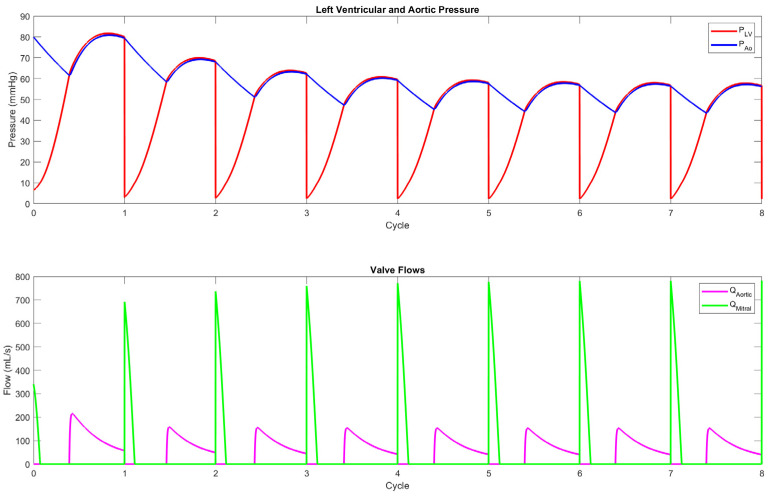
Simulated waveforms generated by a MATLAB-based cMCL model. The top panel shows the pressure waveforms of the left ventricle and aorta, while the bottom panel illustrates the flow rates through the aortic and mitral valves.

**Figure 6 bioengineering-12-00851-f006:**
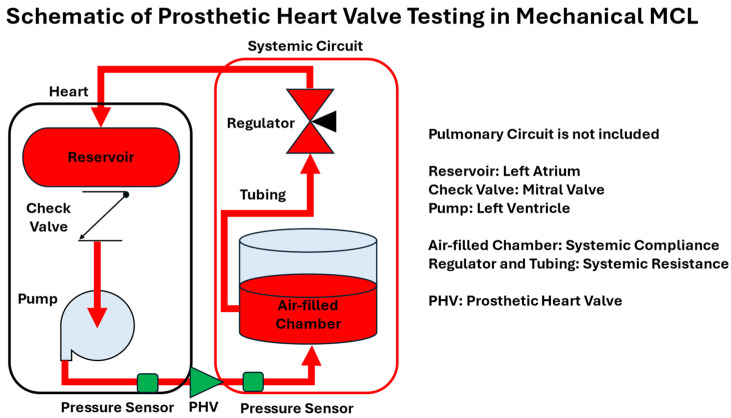
Schematic of prosthetic heart valve testing setup in a Mechanical Mock Circulatory Loop design.

**Figure 7 bioengineering-12-00851-f007:**
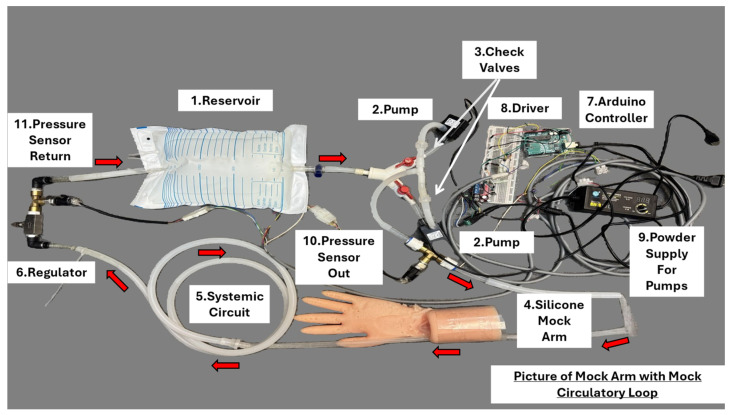
Photograph of the Mock Arm integrated with Mechanical MCL. Fluid begins in the (1) reservoir and is driven by two (2) pumps, each connected to a (3) check valve to maintain unidirectional flow. The fluid then passes through the (4) silicone mock arm tubing and additional tubing that constitutes the (5) systemic circuit. A (6) flow regulator adjusts the return flow to the reservoir, completing the loop. The system is operated via an (7) Arduino controller, which interfaces with the (8) pump driver and (9) power supply to control the pumps. Two pressure sensors, (10) pressure sensor out and (11) pressure sensor return, monitor pressure conditions and provide real-time feedback to the Arduino controller for closed-loop regulation.

**Figure 8 bioengineering-12-00851-f008:**
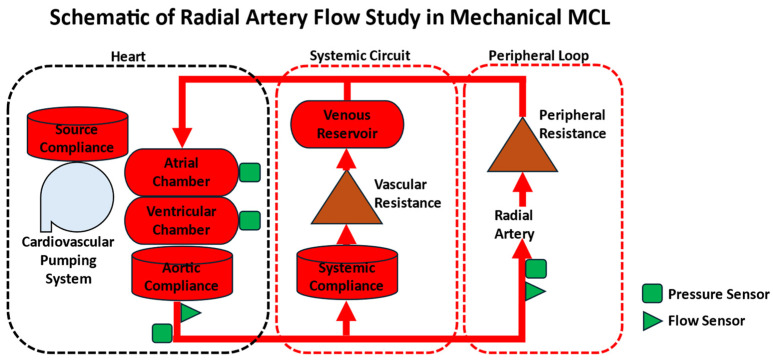
Schematic of the Mechanical Mock Circulatory Loop experimental setup for radial artery flow study.

**Figure 9 bioengineering-12-00851-f009:**
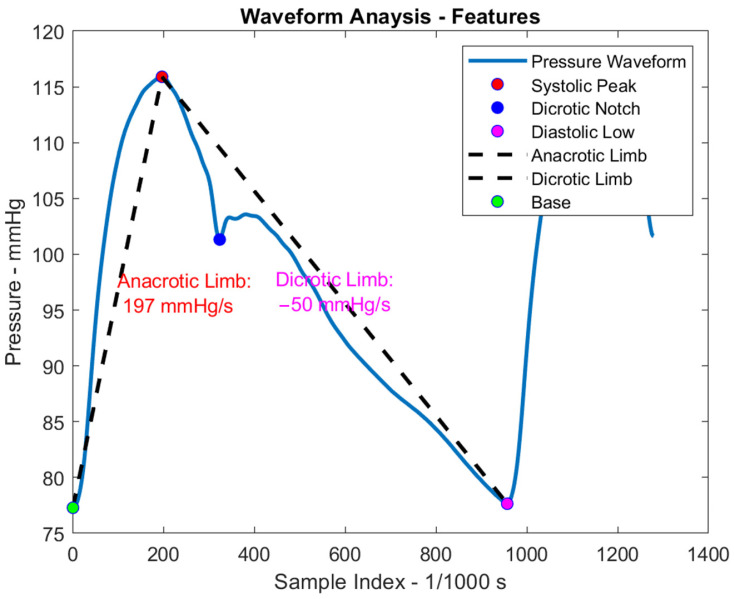
Graph of pressure waveform with features analysis. This figure demonstrates the extraction of key features from a typical brachial artery pressure waveform using MATLAB. Identified features include the systolic peak, dicrotic notch, diastolic low, as well as the anacrotic and dicrotic limbs.

**Table 1 bioengineering-12-00851-t001:** Comparison of the three main types of MCL models.

	Mechanical Model	Computational Model	Hybrid Model
Pros	Realistic fluid behavior	High flexibility	Best of both worlds
	Device testing compatibility	Cost-effective	Patient-specific modeling
	Tangible interface	Rapid simulation	Devic integration
	Standard compliance	Parameter analysis	Closed-loop feedback
Cons	Limited flexibility	No physical interaction	Higher complexity
	Bulky and complex	Requires validation	Cost and setup
	Less adaptive	Software limitations	Requires technical expertise

**Table 2 bioengineering-12-00851-t002:** Summary of MCL applications categorized into four major areas.

Cardiovascular Devices Testing	Clinical Training and Education	Hemodynamics and Blood Flow Studies	Disease Modeling
Ventricular Assist Device	Cardiac Surgery	Hemodynamics of Cardiovascular Conditions	Heart Failure and Cardiogenic Shock
Prosthetic Heart Valve	Catheterization and Intervention Surgical Simulation	Flow of Intra-Aortic Balloon Pump	Hypertension and Peripheral Resistance
Stent, Graft, and Vascular Implant	Medical and Palpation Education	Venoarterial Extracorporeal Membrane Oxygenation	Waveform Analysis
Extracorporeal Circuit, Blood Pump, and Artificial Organ	Perfusion and Extracorporeal Life Support Training	Pulmonary and Systemic Pressure–Flow	Aortic Stenosis and Valvular Performance
Catheter and Guidewire		Pressure, Flow Rate, Vascular Resistance, Compliance	Aneurysm and Stroke

**Table 3 bioengineering-12-00851-t003:** Summary of typical applications for the three main types of MCL models.

Mechanical Model	Computational Model	Hybrid Model
Cardiovascular device testing	Disease modeling	Cardiovascular device testing
Clinical training and education	Device design and optimization	Patient-specific simulations
Waveform analysis	Hemodynamics and flow simulations	Advanced control algorithm development
In vitro clinical trials	In silico clinical trials	Preclinical testing under varied conditions

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
