# Peer review of "Review of Cardiovascular Mock Circulatory Loop Designs and Applications"

_bioengineering, 2025, doi:10.3390/bioengineering12080851_

Round 1

Reviewer 1 Report (Previous Reviewer 1)

Comments and Suggestions for Authors

no further comments

Author Response

Reviewer1

It is quite a comprehensive and timely review on the subject.

It is well written and easy to follow.

Here are some additional comments and suggestions.

Page 1, line 40: replace with “...that have utilised MCLs to...”

Page 4, line 138-139: it would be more appropriate to say “Vozenilek et al. [10] built an advanced MCL which incorporated four pumps and...”

Response: Thank you for pointing this out.  We agree with this comment.  Therefore, we have made the changes.  The revised and edited are in Page 1, line 40 and page 5, line 173.

Reviewer 2 Report (Previous Reviewer 3)

Comments and Suggestions for Authors

This version has addressed my comments. I am happy for this version to go ahead.

Author Response

Reviewer2

Congratulations on creating a comprehensive REVIEW of Mock Circulatory Loops. This MCL will be very useful for basic small and large animal studies, as well as for durability testing and performance evaluation of products nearing clinical application with respect to systemic circulation. However, the paper is quite concise and not much concrete information can be gleaned from it. I do not think it is worth publishing at this point.
1)First, the MCLs are grouped into three categories, but at the very least, the chapter on Mechanical MCLs should explain what systems are used in each of the major papers, with figures for each. Then, it is necessary to describe in detail each system's mechanism, pros, cons, and what kind of animal experiments it is used for in a table.
2) The second point, computational modeling, is also too thin to be worth reading. What kind of imaging systems are used, and what kind of diseases (pediatric, adult, vascular, valve, myocardial) are targeted in each study should be clearly stated in a table.

3) The same applies to hybrid.
4) Figs from several references should be used for actual clinical applications.

Response: Agree.  We have, accordingly, revised 1) mechanical MCLs section with more about the design and added a figure to explain the design, 2) computation MCLs section with more about the design and added a figure to explain the design, 3) hybrid MCLs section with more about the design and added a summary table of the 3 models, 4) added figures for each of the major applications.

3.1 Mechanical:

Revised first sentence: Typical mechanical MCLs are composed of components that simulate the heart chambers, cardiac valves, compliance vessels, and vascular resistance as illustrated in Figure 3.  This configuration represents a basic design focusing on systemic circulation only.  In this setup, mechanical pumps - such as centrifugal or peristaltic pumps – are used to simulate the function of the hear ventricles, generating pulsatile flow that mimics the contraction and relaxation phases of the cardiac cycle.  These pumps move a working fluid (usually water or a blood analog) through the system, replicating the output of the left and right ventricles.  Check valves are strategically placed in the circuit to serve as mitral valves and aortic valves, ensuring unidirectional flow and preventing backflow during the pumping cycle.  To replicate the atriums, fluid containers or flexible bags are used, functioning as temporary reservoirs that hold the fluid before it enters the pump.  Systemic compliance is simulated using an air-filled chamber or vertical water column, which dampens pressure and flow fluctuations, much like the elasticity of arteries.  Finally, systemic vascular resistance is controlled via flow regulators or tubing resistive, allowing users to tune pressure and flow to the desired physiological conditions.  All components are connected with flexible, transparent tubing, enabling visual observation, easy configuration changes, and mimic blood vessels elastance. This modular and mechanical design provides a practical and repeatable platform for cardiovascular device testing, educational demonstrations, and basic hemodynamic studies.

3.2 Computational:

Added 2nd paragraph: Typical computational MCLs are composed of numerical models that represent key components of the cardiovascular system, including the aortic valve, left ventricle, aorta, venous system, and systemic resistance, as illustrated in Figure 4.  These models are governed by time-dependent differential equations that simulate the dynamics of flow (Q) and pressure (P) across the system under varying physiological conditions.  The computational framework uses parameters such as resistance (R), compliance (C), and elastance (E) to mimic the mechanical behavior of each model.  By adjusting these parameters over time (t), the model can simulate realistic cardiac cycles, pressure-volume relationships, and hemodynamic response to interventions or pathological conditions.  Computational MCLs offer a flexible and precise approach for studying cardiovascular function and testing medical devices in silico.

3.3 Hybrid:

Hybrid MCLs integrate mechanical components with computational models to enhance the simulation of physiological responses. This combination leverages the physical accuracy of mechanical systems and the adaptability of computational simulations, allowing for more precise replication of cardiovascular conditions.  Hybrid MCLs consist of mechanical subsystem and computational subsystem.  Mechanical subsystem includes physical components similar to Figure 3, however, typical hybrid MCLs will consist of communication or controlling system as computational subsystem.  This computational subsystem can read real-time pressure/flow data from sensors and computes updated values of resistance, compliance, flow regulator, or pump output.  These systems are designed to overcome the limitations of purely mechanical or purely computational loops by combining the realistic fluid dynamics of physical systems with the adaptability and precision of mathematical modeling.  

Added 3.4 before the last paragraph of 3.3

 3.4. Mock Circulatory Loops Design Summary

The comparison of the three main types of MCL models is summarized in Table 1.

Table 1. Comparison of the three main types of MCL models.

Mechanical Model

Computational Model

Hybrid Model

Pros

Realistic fluid behavior

High flexibility

Best of both worlds

Device testing compatibility

Cost-effective

Patient-specific modeling

Tangible interface

Rapid simulation

Devic integration

Standard compliance

Parameter analysis

Closed-loop feedback

Cons

Limited flexibility

No physical interaction

Higher complexity

Bulky and complex

Requires validation

Cost and setup

Less adaptive

Software limitations

Requires technical expertise

4.1. Mechanical and prosthetic heart valves (PHVs) are another common type of cardiovascular devices tested by MCLs.  A typical setup involves integrating the PHV into the MCL and placing pressure sensors upstream and downstream of the valve, as illustrated in Figure 5.   This configuration enables researchers to adjust hemodynamic conditions within the MCL and monitor resulting changes in pressure and/pr flow, providing valuable data for evaluating the performance of PHVs under various physiological scenarios.

4.2. In regions where traditional medicine practices, such as palpation, are prevalent, MCLs serve as effective training tools. The practice of palpation is discussed in the first chapter of Physical Examination of the Heart and Circulation, a book written by Dr. Perloff [51].  Jeong et al. [52] employed MCLs to simulate cardiovascular physiology, aiding medical practitioners in honing their diagnostic skills through palpation. This application underscores the versatility of MCLs in accommodating various diagnostic techniques across different medical traditions.  A similar setup is illustrated in Figure 6, where a mechanical MCL is connected to a silicone-molded phantom arm.  Silicone tubing is embedded within the phantom to mimic the radial artery, allowing for palpation training under controlled hemodynamic conditions.   

4.3. MCLs are invaluable tools for in vitro studies of hemodynamics, enabling researchers to simulate and analyze cardiovascular conditions such as hypertension, atherosclerosis, and aneurysms. By replicating physiological blood flow and pressure conditions, MCLs provide a controlled environment to investigate the complexities of these diseases without the immediate need for animal or human subjects.  By mimicking the cardiovascular system’s dynamic behaviors MCLs enable the simulation of various pathological conditions – such as hypertension (by increasing systemic resistance), aortic stenosis (by introducing narrowed valve elements), and heart failure (by reducing pump output or ventricular compliance).   These controlled simulations allow researchers to quantify changes in flow patterns, pressure-volume relationships, and shear stress under specific hemodynamic conditions.

Packy et al. [55] investigated the hemodynamic profiles generated by MCLs across three physiological  states: normovolemia, cardiogenic shock, and hyperdynamic circulation.  Their experimental setup, shown in Figure 7, incorporated multiple pressure and flow sensors for comprehensive monitoring.  In another study, Franchini et al. [56] conducted specialized studies on viscoelasticity of human descending thoracic aortas within an MCL.  Barua et al. [57] utilized MCLs to examine cardiac hemodynamics and renal flow concerning intra-aortic balloon pump (IABP) and a novel intra-aortic entrainment pump (IAEP).  Additionally, Schwarzel et al. [58] employed MCLs to study Extracorporeal Carbon Dioxide Removal (ECCO2R) rate without relying on animal models. Stephens et al. [59] and Rozencwaig et al. [60] utilized MCLs to investigate Venoarterial ECMO, while Mi et al. [61] incorporated sound sensors in MCLs to study the pulmonary artery pressure and heart valve vibration sounds.  Papaioannou et al. [62] performed studies on ventricular dysfunction’s effect on arterial compliance using MCLs, and Hong et al. [63] combined commercial pulsatile pumps and 3D printed aortoiliac arterial models connected to MCLs to characterize hemodynamics in human systemic arteries with stenosis.  Koenig et al. [64] explored pressure and volume responses to continuous and pulsatile ventricular assist in MCLs.  Li et al. [65] conducted hemodynamic evaluation of a patient-specific aortic phantom integrated into  a MCL, utilizing pressure sensors and ultrasonic flow sensors to assess flow dynamics and pressure distributions.

4.4. MCLs are powerful tools for cardiovascular disease modeling, enabling researchers to replicate specific pathological conditions and study their hemodynamic impact in a controlled and repeatable environment.  Common cardiovascular diseases - such as heart failure, hypertension, aortic stenosis, aneurysms, and stroke – can be modeled in MCLs by adjusting parameters like loop resistance, tubing compliance, pump output, and valve functionality. 

MCLs have also been applied in waveform analysis and diagnostics, sometimes incorporating machine learning techniques for enhanced pattern recognition and classification.  As illustrated in Figure 8, a normal brachial waveform – sourced from the database created by Willemet et al. [68] - is analysed using MATLAB to extract key morphological features.  These include the systolic peak, dicrotic notch, diastolic low, as well as the anacrotic and dicrotic limbs.  Such analysis enables the quantification of arterial characteristics and serves as a foundation for diagnostic algorithms or machine learning techniques that can identify abnormalities in vascular tone or cardiac function.

Reviewer 3 Report (Previous Reviewer 4)

Comments and Suggestions for Authors

The authors of the manuscript entitled “Review of Cardiovascular Mock Circulatory Loops Designs and Applications” have reviewed and categorized Mock Circulatory Loops (MCLs), emphasizing the need for standardized, simplified, and AI-integrated systems to enhance their utility across biomedical research and clinical applications.

The Introduction section requires stronger support from the literature. Currently, it reads as a series of author assertions. For instance, lines 50–66 should be backed by appropriate references.

The section on computational MCLs needs further elaboration. A flowchart illustrating the process would be more informative than simply listing software packages. Additionally, Figure 4 does not adequately cover this and its quality is suboptimal.

Figure 6 is cluttered and difficult to interpret. Please improve its clarity and presentation.

In Figure 8, the pressure waveform raises questions: Why is the systolic pressure above 140 mmHg, and why is the anacrotic limb slope 250 mmHg/s? These values require justification or correction.

Overall, the manuscript needs thorough proofreading. While it appears to have undergone revisions, it remains too brief to be considered a comprehensive review paper.

Author Response

Reviewer3

This manuscript presented a very basic general summary about the Cardiovascular Mock Circulatory Loops. Although four applications of MCL and three types of MCL were introduced but more details of applications will need to be covered in depth, such as, implementation of stents, how was diseases tested, how will be arterial waves form to be measured and how the system will have the impact on the waveform etc. The summary was too simple which is similar to a training/teaching materials for students. 

Response: Thank you for pointing this out.  We agree with this comment.  Therefore, we have revised each application section and the summary section.

Revised summary section and added a table summary:

1st Paragraph

Based on this literature review, there are many applications for MCLs [6,79].  Four major MCL applications are summarized in Table 2.  The majority of the MCL applications involve testing commonly used cardiovascular devices [80], such as VADs and PHVs.  MCLs have become indispensable in cardiovascular research, from heart valve testing to VAD optimization.  However, there is limited research on the application of MCLs in pediatrics, and their use for clinical training and teaching is also very limited.  Hong et al. [82] recently conducted a comprehensive review of the MCLs applications, with a particular focus on their roles in surgical planning and disease modeling.   In this research, we extensively examine various MCL designs and review different types of applications.  When multiple papers address very similar applications, only representative studies are cited.  Although the applications may be similar, the MCL designs often differ. Given their distinct characteristics, different types of MCL models are often more suitable for particular applications.  Typical applications for the three main types of MCL models are summarized in Table 3.  

Table 3. Summary of typical applications for the three main types of MCL models.

Mechanical Model

Computational Model

Hybrid Model

Cardiovascular device testing

Disease modeling

Cardiovascular device testing

Clinical training and education

Device design and optimization

Patient-specific simulations

Waveform analysis

Hemodynamics and flow simulations

Advanced control algorithm development

In vitro clinical trials

In silico clinical trials

Preclinical testing under varied conditions

Last paragraph:

Cardiovascular waveform analysis [55,71] is another area that requires further development.  With advances in computational modeling, researchers can develop applications for waveform analysis and diagnostics.  As discussed in the context of disease modeling, feature-based analysis is a key method in waveform evaluation.  With combined with machine learning algorithms, this approach can enable the classification of vascular types and the detection of abnormalities in cardiac function.  Furthermore, such feature-based analysis may be valuable for validating the physiological fidelity of MCLs.  Additionally, the integration of MCLs into personalized medicine and in vitro drug testing is very limited [83].

Round 2

Reviewer 3 Report (Previous Reviewer 4)

Comments and Suggestions for Authors

The authors have not addressed the following points. It appears they uploaded another reviewer's comments by mistake. Therefore, I suggest a second round of major revisions.

My initial questions have not been answered yet, as follows: 

1.The section on computational MCLs needs further elaboration. A flowchart illustrating the process would be more informative than simply listing software packages. Additionally, Figure 4 does not adequately cover this and its quality is suboptimal.

2.Figure 6 is cluttered and difficult to interpret. Please improve its clarity and presentation.

3. In Figure 8, the pressure waveform raises questions: Why is the systolic pressure above 140 mmHg, and why is the anacrotic limb slope 250 mmHg/s? These values require justification or correction.

Author Response

Comments:

The authors of the manuscript entitled “Review of Cardiovascular Mock Circulatory Loops Designs and Applications” have reviewed and categorized Mock Circulatory Loops (MCLs), emphasizing the need for standardized, simplified, and AI-integrated systems to enhance their utility across biomedical research and clinical applications.

  1. The Introduction section requires stronger support from the literature. Currently, it reads as a series of author assertions. For instance, lines 50–66 should be backed by appropriate references.

Thank you for pointing this out.  We agree that the Introduction section would benefit from stronger support through citations.  The original text was intended to provide a high-level overview, and some of the references were indeed cited in the later sections.  However, in response to your comment, we have revised the Introduction to include appropriate references previously used in later sections have been moved forward, and additional relevant literature has been incorporated to provide a more robust and well-supported foundation.

  1. The section on computational MCLs needs further elaboration. A flowchart illustrating the process would be more informative than simply listing software packages. Additionally, Figure 4 does not adequately cover this and its quality is suboptimal.

Thank you for pointing this out.  We agree with this comment and have added a figure showing a pressure waveform generated by computational MCL using a Matlab program.  We hope this figure better illustrates the design of the cMCL and enhances the overall clarity of the manuscript.

  1. Figure 6 is cluttered and difficult to interpret. Please improve its clarity and presentation.

Thank you for pointing this out.  We agree that the original version of Figure 6 was overly cluttered and could be difficult to interpret. In response, we have revised the figure to enhance its clarity and presentation. This includes simplifying the layout, improving label visibility, and refining the graphical elements for better readability. Additionally, we have updated the figure caption to provide a clearer and more descriptive explanation of the content, ensuring that the figure is more accessible and informative to readers.

  1. In Figure 8, the pressure waveform raises questions: Why is the systolic pressure above 140 mmHg, and why is the anacrotic limb slope 250 mmHg/s? These values require justification or correction.

Thank you for pointing this out.  We agree with your observation regarding the elevated systolic pressure and the anacrotic limb slope in the original version of Figure 8. The waveform was sourced from a public database; however, we acknowledge that the selected example may not have represented typical physiological values. In response, we have revised the figure by selecting a waveform with more representative brachial pressure readings and improved clarity. The updated figure more accurately reflects normal physiological ranges and enhances the interpretability of the extracted features.

  1. Overall, the manuscript needs thorough proofreading. While it appears to have undergone revisions, it remains too brief to be considered a comprehensive review paper.

Thank you for pointing this out.  We agree that the manuscript required further refinement and detail to meet the expectations of a comprehensive review. In response, we have revised the manuscript thoroughly, adding several new paragraphs to expand key sections and improve the overall depth and clarity. Additionally, a co-author with expertise in academic writing has carefully proofread the entire manuscript to ensure improved readability and coherence.  We would also like to clarify that this review is intended to provide a broad overview of the various designs and applications of mock circulatory loops (MCLs) rather than an in-depth technical analysis of each individual application. Our goal is to highlight the diversity of MCL systems and emphasize the need for standardized, modular, and AI-integrated platforms that can enhance their utility in both biomedical research and clinical diagnostics. We appreciate your observation, as it aligns with the central theme of our work and has helped us better articulate this intent in the revised version.

Round 3

Reviewer 3 Report (Previous Reviewer 4)

Comments and Suggestions for Authors

I would like to thank the authors for answering the questions. 

This manuscript is a resubmission of an earlier submission. The following is a list of the peer review reports and author responses from that submission.

Round 1

Reviewer 1 Report

Comments and Suggestions for Authors

It is quite a comprehensive and timely review on the subject.

It is well written and easy to follow.

Here are some additional comments and suggestions.

Page 1, line 40: replace with “...that have utilised MCLs to...”

Page 4, line 138-139: it would be more appropriate to say “Vozenilek et al. [10] built an advanced MCL which incorporated four pumps and...”

Reviewer 2 Report

Comments and Suggestions for Authors

Congratulations on creating a comprehensive REVIEW of Mock Circulatory Loops. This MCL will be very useful for basic small and large animal studies, as well as for durability testing and performance evaluation of products nearing clinical application with respect to systemic circulation. However, the paper is quite concise and not much concrete information can be gleaned from it. I do not think it is worth publishing at this point.
1)First, the MCLs are grouped into three categories, but at the very least, the chapter on Mechanical MCLs should explain what systems are used in each of the major papers, with figures for each. Then, it is necessary to describe in detail each system's mechanism, pros, cons, and what kind of animal experiments it is used for in a table.
2) The second point, computational modeling, is also too thin to be worth reading. What kind of imaging systems are used, and what kind of diseases (pediatric, adult, vascular, valve, myocardial) are targeted in each study should be clearly stated in a table. 3) The same applies to hybrid.
4) Figs from several references should be used for actual clinical applications.

Reviewer 3 Report

Comments and Suggestions for Authors

This manuscript presented a very basic general summary about the Cardiovascular Mock Circulatory Loops. Although four applications of MCL and three types of MCL were introduced but more details of applications will need to be covered in depth, such as, implementation of stents, how was diseases tested, how will be arterial waves form to be measured and how the system will have the impact on the waveform etc. The summary was too simple which is similar to a training/teaching materials for students. 

Reviewer 4 Report

Comments and Suggestions for Authors

The authors of the manuscript entitled “Review of Cardiovascular Mock Circulatory Loop Designs and Applications” presented a review that categorizes Mock Circulatory Loops (MCLs) into four main areas: cardiovascular device testing, clinical training and education, hemodynamics and blood flow studies, and disease modelling.

While the topic is important and relevant, I found the content too brief to be considered a comprehensive review.

The introduction section is particularly short and does not adequately present the current state of the art in this field.

I also did not understand the purpose of Figure 3—its relevance needs to be clarified.

Additionally, I do not recommend including a figure in the conclusion section.